# Expanding the Applicability of an Innovative Laccase TTI in Intelligent Packaging by Adding an Enzyme Inhibitor to Change Its Coloration Kinetics

**DOI:** 10.3390/polym13213646

**Published:** 2021-10-22

**Authors:** Cheng-Xuan Lin, Hao-Hsin Hsu, Yu-Hsuan Chang, Shih-Hsin Chen, Shih-Bin Lin, Shyi-Neng Lou, Hui-Huang Chen

**Affiliations:** 1Department of Food Science, National Ilan University, Shennong Road, Yilan City 26047, Taiwan; tony853213@gmail.com (C.-X.L.); s4457cindy08@gmail.com (H.-H.H.); tomchang1995@gmail.com (Y.-H.C.); sblin@niu.edu.tw (S.-B.L.); snlou@niu.edu.tw (S.-N.L.); 2Institute of Food Science and Technology, National Taiwan University, Roosevelt Road, Taipei City 10617, Taiwan; shchen0422@ntu.edu.tw

**Keywords:** time–temperature indicator, laccase, electrospinning, sodium azide, kinetic, intelligent packaging

## Abstract

Enzymatic time–temperature indicators (TTIs) usually suffer from instability and inefficiency in practical use as food quality indicator during storage. The aim of this study was to address the aforementioned problem by immobilizing laccase on electrospun chitosan fibers to increase the stability and minimize the usage of laccase. The addition of NaN_3_, as and enzyme inhibitor, was intended to extend this laccase TTI coloration rate and activation energy (*Ea*) range, so as to expand the application range of TTIs for evaluating changes in the quality of foods during storage. A two-component time–temperature indicator was prepared by immobilizing laccase on electrospun chitosan fibers as a TTI film, and by using guaiacol solution as a coloration substrate. The color difference of the innovative laccase TTI was discovered to be <3, and visually indistinguishable when OD_500_ reached 3.2; the response reaction time was regarded as the TTI’s coloration endpoint. Enzyme immobilization and the addition of NaN_3_ increased coloration *Km* and reduced coloration *Vmax*. The coloration *Vmax* decreased to 64% when 0.1 mM NaN_3_ was added to the TTI, which exhibited noncompetitive inhibition and a slower coloration rate. Coloration hysteresis appeared in the TTI with NaN_3_, particularly at low temperatures. For TTI coloration, the *Ea* increased to 29.92–66.39 kJ/mol when 15–25 μg/cm^2^ of laccase was immobilized, and the endpoint increased to 11.0–199.5 h when 0–0.10 mM NaN_3_ was added. These modifications expanded the applicability of laccase TTIs in intelligent food packaging.

## 1. Introduction

Intelligent food packaging is attracting extensive interest from researchers and the food industry, because such packaging aims to monitor food quality in real time during storage and feed back information to logistics managers or consumers [1]. A time–temperature indicator (TTI) is an intelligent packaging device that can record and cumulatively reflect the overall influence of temperature and duration on product quality during storage by using obvious coloration and other methods [2]. A TTI device is attached to the outer packaging of food products; with the device, consumers or managers can easily estimate the risk of food quality deterioration during storage, which previously could only be determined by the expiration date on a food packaging label [3]. Therefore, TTIs can be used as an auxiliary strategy alongside expiration dates to judge changes in the quality of food during storage, which can reduce the risk of consuming deteriorated food before the expiration date, and also help to reduce food waste due to nearing the expiration date.

The principle of an enzymatic TTI is either to hydrolyze the substrate to cause pH changes, or to catalyze redox and other reactions to form colored products and cause the TTI to undergo color changes. Enzymatic TTIs are highly sensitive to environmental temperature alterations, and are highly accurate; consequently, enzymatic TTI systems have been widely studied [4]. Kim et al. [5] successfully developed an enzymatic TTI by using laccase, and presented temperature-dependent coloration. The coloration rate and activation energy (*Ea*) can be adjusted by changing the enzyme and substrate concentrations and adding inhibitors, and the laccase-based TTI coloration system can be modified to respond to the deterioration conditions of a specific product [6].

Laccases (benzenediol: oxygen oxidoreductase; EC 1.10.3.2) are dimeric or tetrameric glycoproteins, and belong to the superfamily of multicopper oxidases (Appendix A). Laccases contain four copper atoms per monomer. These copper sites in laccases are categorized into three groups. To perform their catalytic function, laccases rely on Cu atoms that are distributed at three different copper centers (i.e., Type 1, or blue copper; Type 2, or normal copper; and Type 3, or coupled binuclear copper centers) [7]. At the level of the three-dimensional structure, laccases (bacterial, fungal, and plant) are believed to have three sequentially arranged cupredoxin-like domains. Laccases can oxidize phenols and various aromatic compounds. For phenol oxidation, laccase catalysis comprises three major steps: (1) Type-1 Cu reduction by reducing the substrate; (2) internal electron transfer from Type-1 Cu to the Type-2 and Type-3 Cu trinuclear clusters; (3) reduction of oxygen (to water) in Type-2 and Type-3 Cu [8,9]. The resultant phenoxy radicals that are produced during the oxidation of phenolic compounds can undergo various reactions, including the following: radical–radical coupling reactions of monomers for the synthesis of dimers, oligomers, and polymers; radical cross-coupling reactions; and in situ generation of *ortho*-quinones and *para*-quinones from the corresponding catechols and hydroquinones, respectively, via disproportionation [10]. Pigments are then developed after polymerization. The coloration of laccase-catalyzed oxidation of guaiacol (Appendix A) can be expressed as follows:
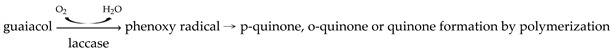
(1)

However, the disadvantages of enzymatic TTI sinclude enzymatic instability, high cost, and a small monitoring range, all of which must be addressed [11]. To reduce the influence of enzymatic instability and the high cost of enzymes, Tsai et al. [12] immobilized laccase on electrospun chitosan fibers to develop an innovative TTI. Electrospinning is a progressive and desirable method that produces fibers with diameters ranging from submicrons to several nanometers in order to obtain materials with a large specific surface area. Recently, bio-based electrospun nanofibers have gained substantial attention for preparing polymer-based biomaterials intended for use in cell cultures [13], multifunctional silk-fibroin-based devices [14], and food packaging [15]. In addition, the electrospun fibrous network can alleviate the three-dimensional obstacles faced by reaction products [16]. The amount of enzyme required is diminished, thereby minimizing TTI production costs [12]. The immobilization of laccase with covalent crosslinking on the electrospun chitosan fibers also results in increased enzyme stability [17].

Nevertheless, the enzymatic TTI that immobilizes laccase on electrospun fibers still has some disadvantages, including the pigmentation appearing after the long coloration duration and a narrow range of coloration *Ea*, affecting its applicability and monitoring range, respectively [12,17]. The former disadvantage might be resolved by adding dispersing substances [18], and the latter might be resolved by adding an enzyme inhibitor [6]. To reduce the influence of the small monitoring range of enzymatic TTIs, numerous inhibitors (especially compounds that can form stable copper complexes) have been used to inhibit laccase. The compounds used include dithiothreitol, thioglycolic acid, cysteine, diethyldithiocarbamic acid, ethylenediaminetetraacetic acid, sodium fluoride, and sodium azide (NaN_3_, Na–N=N^+^=N^−^) [19]. Among them, the reducing reagent NaN_3_ is most frequently used. NaN_3_ inhibits enzyme activity by binding T_2_/T_3_ TNC copper sites and, in turn, blocking the electron transfer and reduction of molecular oxygen in laccases [20,21]; the increased concentration then leads to decreased laccase activity [22].

Fresh-cut fruits and vegetables (FCFVs) in the cold chain are prone to deterioration and microbial contamination. Although the use of active packaging—such as coating the edible film with natural polyphenols on freshly cut apple pieces—can increase the antioxidant activity of FCFVs [23], most of their quality cannot be judged from their appearance. The TTI, prepared by immobilizing laccase on electrospun chitosan fibers, has proven to be useful in monitoring changes in the quality of FCFVs in the cold chain [17]. In the present study, the coloration rate and *Ea* of the laccase TTI, as well as the inhibitory kinetics of NaN_3_, were investigated so as to extend the *Ea* range and coloration duration; moreover, adjustments to the quantities of laccase and NaN_3_ were designed to expand the range of application of laccase TTIs in intelligent packaging for food products, in order to improve the accuracy of the TTI in estimating the changes in the quality of various foods.

## 2. Materials and Methods

### 2.1. Immobilizing Laccase on Electrospun Fibers

Following the method of Tsai et al. [12], a chitosan (CS, DD: 76%, 50–190 kDa, Sigma-Aldrich, St. Louis, MO, USA) composite gel solution with polyvinyl alcohol (PVA, 118–124 kDa, First Chemical Manufacture, Taipei, Taiwan) and tetraethoxysilane (Si(OEt)_4_, Sigma-Aldrich) was electrospun on a polypropylene (PP) film using an electrospinning apparatus (NE-300, Falco Enterprise, New Taipei City, Taiwan). The electrospun fibers were collected on a polypropylene (PP) film, which was attached to the drum collector, 12 cm from the tip of the needle. The completed single-side electrospun film (material 1, M1) was soaked in 3% glutaraldehyde (GA; Nihon Shiyaku Industries, Osaka, Japan) and incubated for 2 h. Subsequently, the GA-modified M1 was washed with acetate buffer (pH 4.5) and dried overnight in a desiccator at ambient temperature to produce M2. A total of three sheets of M2 were produced, and 20 pieces each measuring 1 cm^2^ were cut from the middle section of each sheet. A minimum of three pieces were randomly selected for enzyme immobilization and coloration tests. To perform enzyme immobilization, 15–25 μg of laccase (Sigma-Aldrich, from *Trametes versicolor*, ≥0.5 U/mg, lot result: 1.07 U/mg) in 50 μL of buffer solution was adsorbed carefully onto a 1 cm^2^ piece of M2, and the crosslinking reaction—by which covalent bonds were formed between the amine group of laccases and the aldehyde group of GA—was set for 12 h at 5 °C to immobilize laccase on each piece of the film. The electrospun film–immobilized enzyme was dried in an oven for 1 h at ambient temperature to produce M2-laccase. In addition, the occurrence of a coupling reaction in M2-laccase was tested by rinsing it with a guaiacol solution to verify that the laccase was effectively immobilized.

### 2.2. Guaiacol Solution Preparation

Fish gelatin, which was purchased from Jellice Pioneer (Pingtung, Taiwan), was prepared from tilapia skin, and had a bloom value of 200 g. The gelatin was dissolved in 1 mM acetic acid–sodium acetate buffer solution (pH = 4.5) and heated for 3 h at 121 °C to form 2.5% hydrolyzed gelatin (HG). Guaiacol (Acros Organics, Morris Plains, NJ, USA) was dissolved in the HG to form 20 mM guaiacol-added HG (HGG). NaN_3_ (Merck Millipore, Kenilworth, NJ, USA) was dissolved in HGG to form 0–0.10 mM NaN_3_-added HGG (N-HGG) as a coloration substrate solution.

### 2.3. Coloration of Laccase TTI Prototype

The 1 cm^2^ pieces of M2-laccase with varying amounts of immobilized laccase (15–25 μg/cm^2^) were immersed in 1 mL of N-HGG solution to obtain the laccase TTI prototype, and to investigate its coloration.

#### 2.3.1. Color Measurement

The color of the laccase TTI prototype was measured using six replicates with a color measurement spectrophotometer (Diffuse/8° Spectrophotometer, Hunter Associates Laboratory, Reston, VA, USA). The results were expressed in accordance with the CIELAB system. The parameters determined were the degrees of lightness (*L**), redness (*+a**) or greenness (−*a**), and yellowness (*+b**) or blueness (−*b**). The color difference (Δ*E*) between the samples before (the sample with subscript 0 in Equation (1) and after coloration of the laccase TTI prototype was calculated as follows:(2)$$\Delta E=\sqrt{{\left(L^{*}{}_{1}-L^{*}{}_{0}\right)}^{2}+{\left(a^{*}{}_{1}-a^{*}{}_{0}\right)}^{2}+{\left(b^{*}{}_{1}-b^{*}{}_{0}\right)}^{2}}$$

#### 2.3.2. Absorbance of Coloration

The spectrophotometer can only determine the color of a single sample with each measurement; therefore, conducting multiple experiments may result in differences in time and interference in the coloration data. Therefore, an ELISA reader (BioTek Instruments, Winooski, VT, USA) was used to simultaneously detect the coloration of multiple samples. The guaiacol oxidation in N-HGG was measured based on the increase in absorbance at 500 nm. According to Jayalakshmi and Santhakumaran (2011) [24], the normalized absorbance (Abs_norm_) was calculated as the OD_500_ of the sample divided by that of the coloration endpoint. The coloration endpoint was a Δ*E* of <3, even when the reaction time was extended [25].

### 2.4. Kinetic Analysis of Laccase TTI Prototype Coloration

#### 2.4.1. Enzyme Kinetics

As described by Qiu et al. [26], 20 µg of the free laccase or the immobilized laccase on M2 was applied to react with a set of virtual substrate concentration ([S]) data, without considering residual errors, which was simulated for four initial guaiacol concentrations of 1, 0.5, 0.25, and 0.125 mM in HGG with or without NaN_3_ at a prespecified time interval. The maximum reaction velocity (*V_max_*) and the Michaelis–Menten constant (*K_m_*) were estimated using Lineweaver–Burk plotting, as follows:(3)$$\frac{1}{V}=\frac{k_{m}}{V_{max}\left[S\right]}+\frac{1}{V_{max}}$$

#### 2.4.2. Coloration Rate

Laccase TTI prototypes were incubated at 5, 15, 25, and 35 °C (the standard deviation of the actual temperature was 1 °C). The coloration rate was calculated as the change in OD_500_ within a particular time interval. The color response of TTIs, plotted as a function of time, followed the sigmoid pattern described by the following logistic Equation (4):(4)$$y=\frac{1}{1+\exp\left(\frac{k_{1}-t}{k_{2}}\right)}$$
where *y* is the OD_500_ determined at coloration time *t*, and *k*_1_ and *k*_2_ are the response rate constants serving as the functions of enzyme concentration and storage temperature, respectively. Note that 1/4*k*_2_ is the exponential rate constant—that is, the phase’s slope *k* in which the TTI’s response changes exponentially with time. Each temperature value of *k*_1_ and *k*_2_ was determined using nonlinear regression analysis (SigmaPlot 10.0, Systat Software, Chicago, IL, USA).

#### 2.4.3. Activation Energies

*Ea* (kJ/mol) was calculated according to the Arrhenius expression:(5)lnk=−Ea/RT+lnA
where *R* is the general gas constant (8.314 J/K·mol), *T* is the absolute temperature (K), and *A* is the prefactor. The *Ea* of the laccase TTI prototype was estimated by using the slope of the Arrhenius plot. *Ea* was estimated at least thrice, and the coefficient of determination (R^2^) was calculated for each test. The highest R^2^ was selected to represent the accuracy of *Ea* estimation.

#### 2.4.4. Dynamic Temperature Response Test

Following the method of Tsai et al. [12], temperature fluctuations between 4 °C (refrigeration temperature) and 25 °C (room temperature) were applied to simulate dynamic storage conditions. Incubation of the laccase TTI prototype at 4 °C for 8 h, followed by storage at 25 °C for 8 h, was performed three times. After each incubation of 8 h, the OD_500_ of the TTI prototype was quickly determined, and it was then incubated again at the other temperature.

### 2.5. Statistical Analysis

All treatments were performed at least in triplicate. The data were analyzed using SAS (SAS 9.4, Cary, NC, USA). All statistical analyses were conducted in triplicate, and the results were expressed as the mean and standard deviation. The average values were compared using ANOVA at the least significance level *p* < 0.05.

## 3. Results and Discussion

### 3.1. Coloration of Laccase TTI Prototype

The morphologies of electrospun CS/PVA/Si(OEt)_4_ fibers on PP film with diameters evenly and mainly distributed around approximately 200 nm were observed (Figure 1). The uniform and fine electrospun chitosan fibers constitute a desirable carrier with a large specific surface area for the immobilization of laccase [17]. The laccase TTI prototype changed color from reddish brown to light amber with precipitation after undergoing 2 h of coloration at 25 °C (Figure 2A). Laccase catalyzes the oxidation of the phenolic hydroxyl group on the aromatic monophenol structure of guaiacol to generate quinone compounds and present color [5]. Thereafter, laccase could polymerize into dimers, trimers, or larger polymers [27]. In the present study, uneven color changes could be observed periodically after coloration in the TTI prototype had occurred for some time; this phenomenon was speculated to have resulted from the precipitation caused by increased polymerization of quinone compounds. Consequently, the TTI prototype formed a barrier on the surface of the laccase due to the precipitation of particles stacked on the electrospun fiber; this affected the contact between the laccase and guaiacol, and interfered with the catalytic oxidation reaction, leading to unstable coloration in the TTI prototype.

In general, a TTI is applied to monitor the condition of packaged foods and their surrounding environments, and it can also provide visual monitoring via color changes that correspond to accumulated time and temperature information [28]. In order to enable managers and consumers to accurately evaluate information on changes in food quality, dispersing substances were added to the guaiacol solution in the following experiment to increase the uniformity of the TTI prototype’s coloration.

Wang and Li [18] found that adding gelatin can improve the oxidation stability of guaiacol. However, the gelation of gelatin occurs at low temperatures. Since the heated fish skin gelatin with weak gel strength at 121 °C for 3 h formed HG that was fluid at refrigeration temperature [29], the HG was then used as the dispersion substance in TTI prototype coloration. The results of the present study indicate that no precipitation was generated and uniform color change was achieved during TTI prototype coloration when HG was added to the guaiacol solution (Figure 2A); the maximum absorption peak of 500 nm was stably achieved in the spectrum.

The laccase TTI prototype changed in color from transparent to light brown, dark brown, reddish brown, reddish purple and, finally, purplish brown; Δ*E* exhibited a nonlinear positive correlation with OD_500_ during the coloration process (Figure 2B). At the initial stage of coloration, Δ*E* was larger, and indicated clearer color changes. Thus, the color of the TTI prototype gradually deepened during isothermal coloration. The less the change to Δ*E*, the more difficult it was to distinguish the color change. The coloration endpoint was defined as the minimum time taken for Δ*E* to decrease to less than 3, even when the reaction time was extended; at this point, color changes to the laccase TTI prototype could not be distinguished visually [25]. Past this point, measuring the absorbance value was meaningless, given that the TTI is intended for commercial use as a smart packaging [30]. Therefore, in the present study, the OD_500_ (3.2) at which the corresponding Δ*E* was less than 3 (compared with the deepest TTI color) was regarded as the coloration endpoint of the laccase TTI. Abs_norm_ was calculated by dividing the OD_500_ of the sample by 3.2.

### 3.2. Coloration of Laccase TTI Prototype with NaN_3_

To accurately predict the deterioration in the quality of various foods during storage, the amounts of immobilized laccase and added NaN_3_ were adjusted to regulate the coloration rate and *Ea* of the laccase TTI prototype. Coloration lag was observed at the simulated cold chain temperature of 5 °C (Figure 2). In other words, no colors were initially observed, and the color that appeared was markedly deepened prior to approaching the coloration endpoint (Appendix A). When more NaN_3_ was added or less laccase was immobilized, the coloration lag phase was extended. This finding indicates that when the ratio of enzyme inhibitor to immobilized enzyme was higher, the effect of hysteretic coloration was more pronounced, and the time to reach the coloration endpoint in the laccase TTI prototype was longer. For example, in 0.1 mM N-HGG (representing the concentration of NaN_3_ in HGG is 0.1 mM), the time to reach the coloration endpoints of TTI prototypes with 15, 20, and 25 μg/cm^2^ of immobilized laccase at 5 °C was increased from 21.8 ± 0.6 h to 199.5 ± 2.1 h, from 14.4 ± 0.4 h to 189.8 ± 0.8 h, and from 11.0 ± 0.8 h to 167.7 ± 0.2 h, respectively (Appendix A).

In the other form of the fungal laccases, the four copper ions were in the +2 oxidation state. The catalytic cycle began with the sequestering of one electron at a time from the reducing substrate, and the transfer of the electron to the trinuclear T2/T3 copper cluster through a highly conserved His–Cys–His pathway [8]. During this process, the NaN_3_ halted substrate oxidation by preventing O_2_ uptake, and by interfering with the electron’s transfer to the trinuclear center during the initial reaction of laccase-catalyzed guaiacol oxidation, thereby inhibiting coloration. After a period of reaction, colored quinone products—which are oxidized from guaiacol—increased in volume, reached a steady state, and gradually approached the endpoint of coloration [31].

When Bobelyn et al. [32] monitored mushrooms using a lipase TTI, the TTI changed from green to yellow, prolonged with storage time, and the hue change also showed an S-shaped trend. Some antioxidants were added to a TTI that was used to monitor quality changes in ready-to-eat sandwiches; the authors discovered that their TTI exhibited a coloration lag and an S-shaped trend for color changes [30]. The color response curve over time of the present study also exhibited a similar sigmoid pattern (Figure 3).

A clearer coloration lag and longer duration to reach the coloration endpoint of the TTI were observed in a low-temperature environment (Figure 4), indicating that NaN_3_ inhibits laccase activity more significantly at low temperatures. Nevertheless, when the ambient temperature rises and deviates from the cold chain temperature, the TTI coloration can still be accelerated. This phenomenon indicates that the TTI is useful for managing food quality in the cold chain. A laccase TTI exhibits almost no color reactions prior to food deterioration when the food product is stored at a low, stable temperature. This helps to alleviate consumers’ concerns about purchasing food products whose TTI color has changed.

### 3.3. Coloration Kinetics of Laccase TTI Prototype with Added NaN_3_

To analyze the coloration kinetics in the free laccase and the laccase immobilized on the TTI prototypes, with or without NaN_3_, a Lineweaver–Burk plot was created (Figure 5). The immobilization changed the *K_m_* of laccase from 1.40 (for free enzymes) to 1.55 mM (for immobilized enzymes), and changed the *V_max_* of laccase from 0.082 mM/min (for free enzymes) to 0.064 mM/min (for immobilized enzymes). An increase in *K_m_* and decrease in *V_max_* were also observed for laccase in the TTI prototype with NaN_3_. The increase in *K_m_* revealed that the decrease in the immobilized enzyme affinity could be correlated with the limitation of enzyme substrate diffusion to the active site, or the increase in enzyme rigidity. The decrease in *V_max_* might have resulted from linkages that restricted the enzyme active site’s availability for substrate, diffusional limitations, or steric effects [33].

However, the immobilized laccase still retained 78% of the *V_max_* of the free laccase, although the immobilization reduced the affinity of the enzyme to the substrate. George and Sugunan [34] found that the immobilization of amylase also increased its *K_m_* and reduced its *V_max_*; they concluded that the active site of the enzyme had changed, and that the diffusion of the substrates made it difficult for them to reach the active site of the enzyme. Zdarta et al. [35] reported that the *V_max_* of immobilized laccase on electrospun fibers produced using the adsorption method was 75% of that of free laccase; this indicates that immobilized laccase is still useful for catalyzing substrates.

The Lineweaver–Burk plot and *V_max_* variation results were evaluated, and NaN_3_ was speculated to behave as a noncompetitive inhibitor in relation to immobilized laccase. *K_m_* was approximately 2.10 mM when 0.1 mM NaN_3_ was added to a laccase TTI prototype, and *V_max_* decreased to 64% of that observed in the prototype without NaN_3_ (Figure 5); this indicates that the TTI with NaN_3_ had a significantly reduced coloration rate.

Furthermore, the sigmoid pattern was applied to evaluate the coloration kinetics of the laccase TTI prototype with NaN_3_, because a lag phase was observed, and similar sigmoid curves were formed during coloration. The laccase TTI prototype without NaN_3_ (0 mM N-HGG) exhibited a higher *k* value at higher temperatures when the quantity of immobilized laccase remained unchanged (Figure 6A–C); this indicates that the laccase TTI prototype was sensitive to temperature. Brizio and Prentice [36] also reported a positive correlation between the coloration rate constant and the enzyme dosage and temperature in an amylase TTI. However, in the present study, the amount of immobilized laccase was adjusted to 15–25 μg/cm^2^ (equal to 0.016−0.027 U/cm^2^), and the *Ea* of 0 mM N-HGG only changed from 29.92 ± 2.20 to 33.27 ± 0.42 kJ/mol (Figure 6D). These results demonstrate the limited effect of expanding the *Ea* range of the laccase TTI prototype when the only change made was to the enzyme concentration. Other studies have reported similar findings: Kim et al. [5] maintained a free laccase dosage to be within 0.104−0.650 activity units, and observed that the resulting coloration *Ea* only changed within the range of 43.90–45.44 kJ/mol; Wu et al. [37] maintained a lipase level at 0.402−0.550 g/100 mL, and observed that the resulting coloration *Ea* only changed within the range of 34.32−42.12 kJ/mol.

The addition of NaN_3_ could notably increase the coloration *Ea* of the laccase TTI prototype, and expanded its range to 29.92 ± 2.20 to 66.39 ± 1.74 kJ/mol (Figure 6D). Since the addition of NaN_3_ can change the redox potential difference of the laccase reaction [38], the addition of N-HGG in a laccase TTI altered the electron transfer of copper clusters, delayed the coloration endpoint, and expanded the *Ea* range.

Park et al. [6] used laccase to oxidize 10 mM ABTS solution to prepare a laccase TTI; they extended the coloration *Ea* range of the laccase TTI to 48−110 kJ/mol by adding 0–0.10 mM NaN_3_. The coloration *Ea* achieved in the aforementioned study was larger than that of the laccase TTI prototype developed in this study. The main differences between the two experiments were related to the application of different coloration substrates and immobilization of the laccase. The coloration was observed as a zero-order reaction by Park et al. [6], and as a sigmoid pattern in the present study. NaN_3_ behaved as a noncompetitive inhibitor that could interfere with the TNC electron transport of laccase and inhibit the oxidation of guaiacol. Nevertheless, the laccase was crosslinked and immobilized on the electrospun chitosan fiber in the present study; therefore, the coloration developed by oxidizing guaiacol to quinones was more stable, and exhibited lower temperature sensitivity; hence, the effect of coloration on *Ea* amplification was not as significant.

In the laccase TTI prototype, the decrease in the coloration rate constant (*k*) did not completely correspond to the increases in added NaN_3_ (Figure 6A–C). The *Ea* peaked for the TTI with 0.05 mM NaN_3_ when the amount of immobilized laccase was 15 μg/cm^2^ (Figure 6D). In a previous study, laccase was used to prepare biofuel cells for use as TTIs, and the researchers observed that the *Ea* increased from 52.756 to 78.222 kJ/mol when 0–0.02 M NaN_3_ was added, but decreased when the amount of NaN_3_ added was greater than 0.02 M [3]. This phenomenon could be related to the limitations imposed on coloration rate interruption and *Ea* amplification when NaN_3_ was saturated while binding with copper ions at the active site of the laccase. This inference is supported by the fact that the coloration *Ea* continued to increase even when the NaN_3_ concentration was increased to greater than 0.05 mM and the immobilized laccase was increased to 25 μg/cm^2^. Therefore, although increasing the amount of immobilized enzyme shortens the time needed to reach the coloration endpoint (Figure 3 and Figure 4), adding more NaN_3_ with a high concentration of immobilized enzyme can effectively increase the coloration *Ea* of the laccase TTI prototype (Figure 6).

The kinetic parameters of the laccase TTI prototypes were further verified under non-isothermal conditions to simulate temperature fluctuations that may occur in a cold chain, in order to determine whether laccase TTI prototype coloration was reproducible in terms of the increase in coloration with an increase in temperature, and then the decrease in coloration with a decrease in temperature. Therefore, a TTI prototype with 25 μg/cm^2^ laccase and 0.075 mM N-HGG was employed in dynamic temperature tests. The results showed that the reaction rate (*k*) of laccase at 4 °C was lower than that at 25 °C in each cycle (Figure 7), indicating that the laccase TTI prototype exhibited a favorable response to temperature, and that the coloration was reproducible under temperature fluctuations. Temperature sensitivity was thus retained by the immobilized laccase during temperature fluctuations, and dynamic modeling performed through kinetic parameters and numerical analysis was effective. Because coloration hysteresis appeared in TTIs with NaN_3_—particularly at low temperatures—the Abs_norm_ increased after 24 h of reaction, and the color began to change at a 0.25/h coloration rate. However, the color of the TTI prototype deepened significantly from the 40th hour, and then quickly reached the coloration endpoint at 25 °C. Whether a TTI is suitable to monitor a certain product depends on both its temperature dependence under isothermal and non-isothermal conditions, and on the underlying kinetic mechanism [32]. The laccase TTI with NaN_3_ met the aforementioned conditions, and revealed that this TTI can be successfully applied to estimate changes in food quality during storage.

## 4. Conclusions

Enzymatic TTIs were successfully prepared by immobilizing laccase on electrospun chitosan fibers. The desirable tolerance and sensitivity to temperature fluctuations of laccase TTIs with NaN_3_ is an important contribution of intelligent packaging to provide information on temperature alternation and time accumulation during food storage, reflecting changes in food quality. This laccase TTI can be useful to monitor changes in the quality of foods during storage, and to make up for the unreliability of judging the shelf life based on the expiration date on the package. Moreover, the *Ea* and coloration rate of this laccase TTI prototype can be easily adjusted by changing the amount of immobilized laccase, and by adding NaN_3_ to expand the provision of accumulated information on temperature and time during food storage. Subsequently, the prototype’s scope of application in intelligent food packaging can be expanded. When an enzyme inhibitor was added to the TTI prototype, another feature that was gained was the hysteresis of coloration and the absence of color in the initial reaction stage. A noticeable color deepening was observed near the coloration endpoint. Consumers may be less willing to purchase a product if the color of a product’s TTI appears too early, believing that the product has been left on the shelf for a long time. A design in which enzyme inhibitors are added can alleviate this problem and improve the commercial viability and applicability of TTIs.

## Figures and Tables

**Figure 1 polymers-13-03646-f001:**
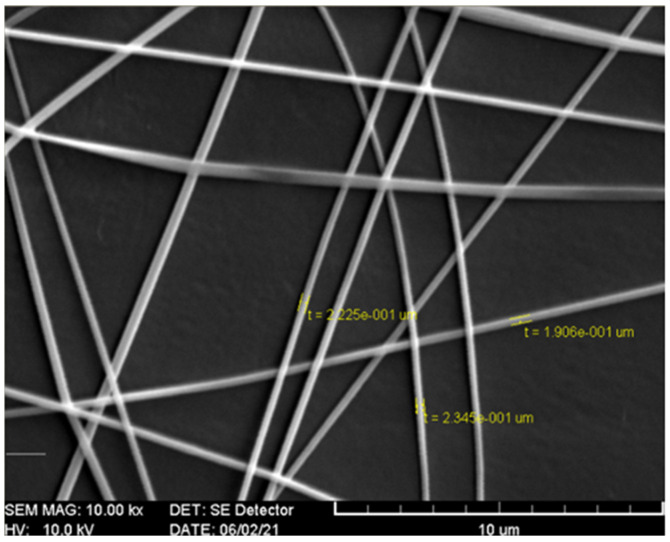
Scanning electron microscopy images of electrospun chitosan/polyvinyl alcohol/tetraethyl orthosilicate fibers on polypropylene film.

**Figure 2 polymers-13-03646-f002:**
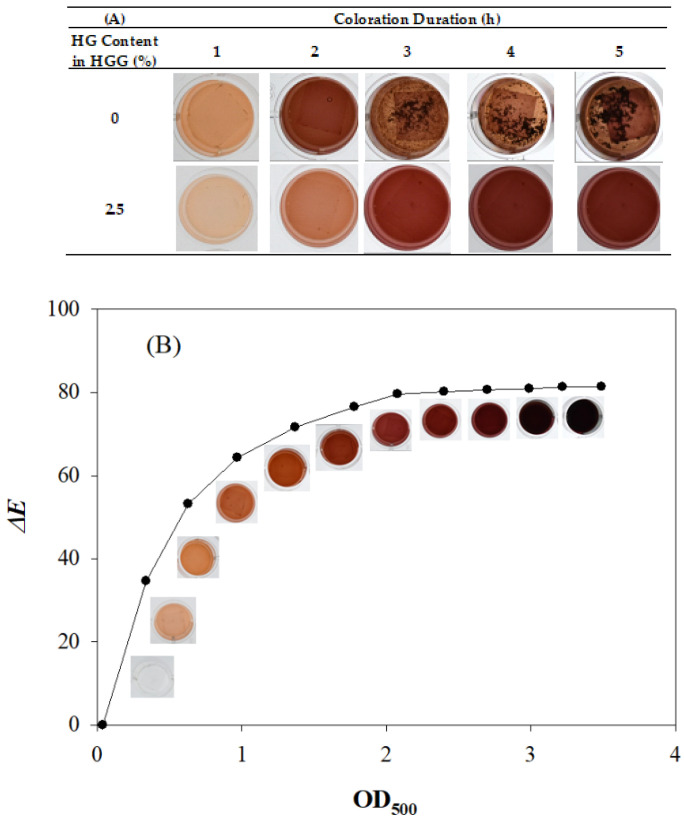
(**A**) Color change of guaiacol solution with and without 2.5% hydrolyzed gelatin (HGG) reacting with M2-laccase (20 μg/cm^2^ laccase) at 25 °C. (**B**) Relationship between Δ*E* and OD_500_ resulting from the coloration of the laccase TTI prototype (HGG reacting with M2-laccase immobilizing 20 μg/cm^2^ laccase).

**Figure 3 polymers-13-03646-f003:**
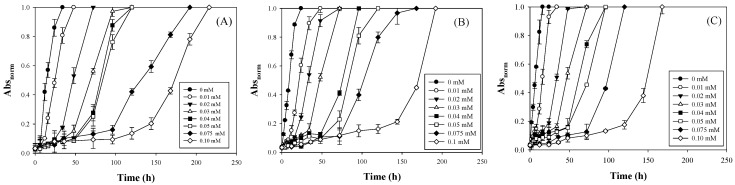
Coloration of TTI prototype containing (**A**) 15, (**B**) 20, and (**C**) 25 μg/cm^2^ of laccase and 0–0.1 mM N-HGG during isothermal storage under 5 °C.

**Figure 4 polymers-13-03646-f004:**
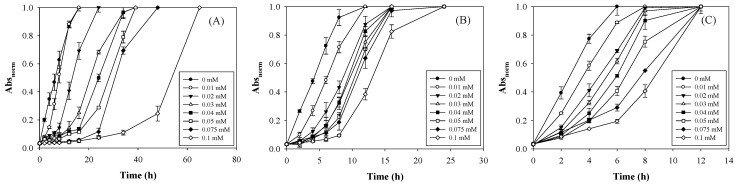
Abs_norm_ changes of a TTI prototype with immobilized laccase of 20 μg/cm^2^ and 0–0.10 mM N-HGG under (**A**) 15, (**B**) 25, (**C**) and 35 °C.

**Figure 5 polymers-13-03646-f005:**
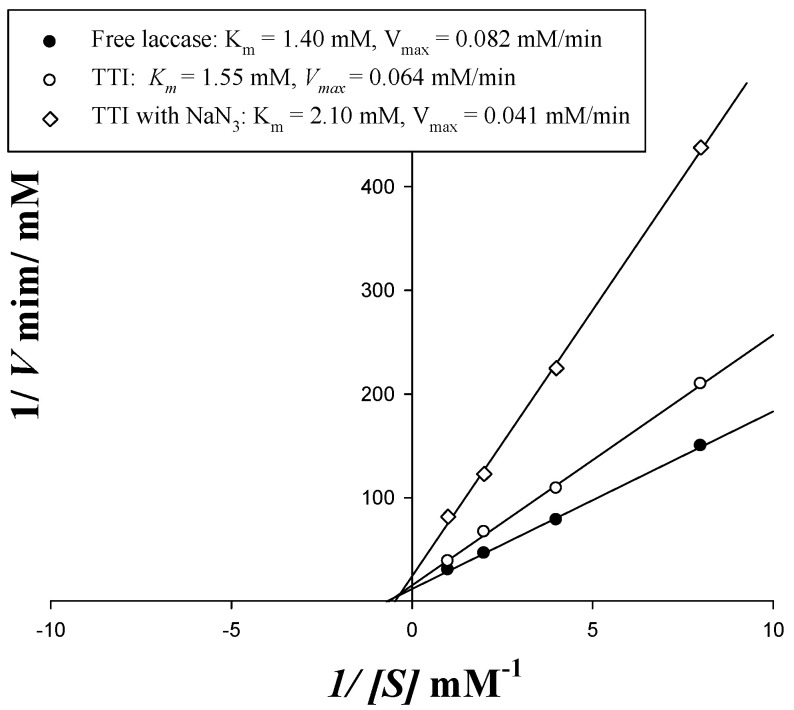
Kinetic parameters of free laccase and a TTI prototype with 20 μg/cm^2^ immobilized laccase (with and without 0.10 mM N-HGG).

**Figure 6 polymers-13-03646-f006:**
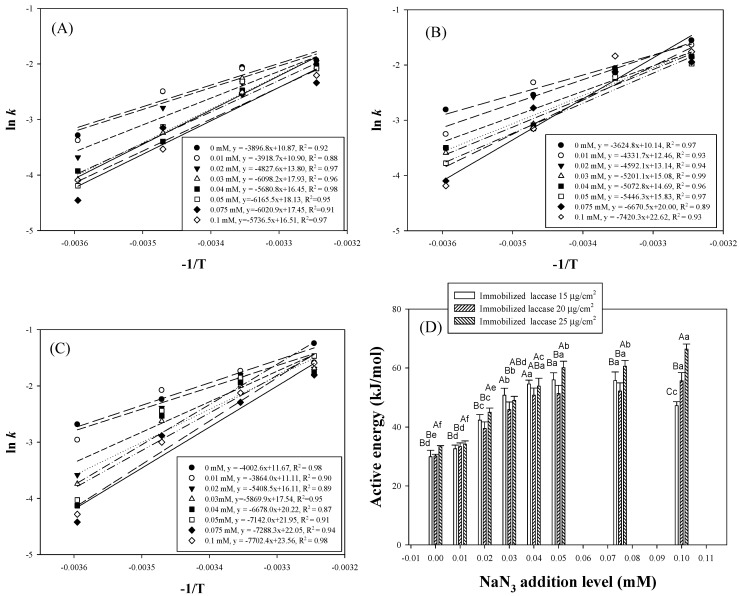
Coloration rate constant of TTI prototypes containing (**A**) 15, (**B**) 20, and (**C**) 25 μg/cm^2^ laccase, 0–0.1 mM N-HGG, and 0–0.1 mM N-HGG; (**D**) Arrhenius activation energy (*Ea*) of the laccase TTI prototype. (**A**–**C**): significant difference in *Ea* with respect to the quantity of enzyme immobilized; a–f: significant difference in *Ea* with respect to the amount of NaN_3_ added (*p* < 0.05).

**Figure 7 polymers-13-03646-f007:**
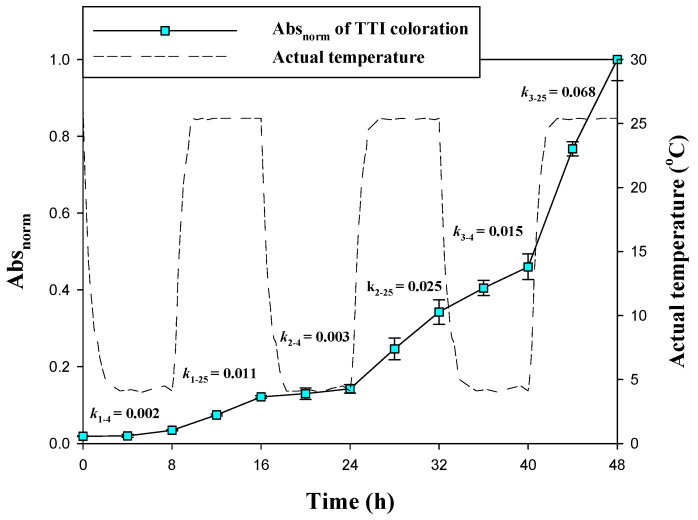
The Abs_norm_ of the TTI prototype onto which 25 μg/cm^2^ laccase was immobilized and 0.075 mM N-HGG was added (*k* is the coloration rate, Abs_norm_/h; the 1st and 2nd subscripts of *k* represent the number of cycles and temperature, respectively) under dynamic temperature conditions: three cycles of 8 h each at 4 and 25 °C.

## Data Availability

Data are available on request from the corresponding author.

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
