# Peer review of "Expanding the Applicability of an Innovative Laccase TTI in Intelligent Packaging by Adding an Enzyme Inhibitor to Change Its Coloration Kinetics"

_polymers, 2021, doi:10.3390/polym13213646_

Round 1
Reviewer 1 Report
The paper looks like some technical description of tests done for specifical application and lacks scientific soundness. The amount of multi-letter non-universal abbreviations makes reading and understanding impossible. Lack of scientific explanation (units, concentrations, conditions), the authors didn't care if anybody could reproduce the results. Jargon-like language did not make reading easier. Results are of great significance for the food storage - multi-billion value element of a food supply chain, yet the authors gave very few hints, what they did.
The system is based on the nanofibrous membrane attached to unclear polypropylene material (mesh? membrane? film?). There is no single electron microscopy image proving that there were nanofibers (rather micro and nanofibers). The "spreading" of the enzyme is not an immobilization, and produced material may have a practical value, yet the enzyme is not immobilized - it's still spread. No single chemical structure is presented, no scheme of the reaction. The idea of the colors' saturation is not connected to any literature, so no source is given for the majority of the measurements. Table 1. could find its' place in the supplementary material. Saved space can be used for more transparent writing. The two sentences: "The electrospun film–immobilized enzyme was dried in an oven for one h at ambient temperature to produce CS/PVA/TEOS/PP/GA/laccase (CPTPGL). In addition, the occurrence of a coupling reaction in CPTPGL was tested by rinsing CPTPGL with a guaiacol solution to verify that the laccase present on each batch of CPTPGL was effectively immobilized." are entirely incomprehensible due to the overuse of often recurring multi-letter abbreviations.
Basing on these severe faults, I do recommend to re-write the paper and encourage re-submission
Author Response
Response 1:
Appreciate the suggestions from the reviewer. The authors have rewritten part of the narrative in the manuscript to improve the readability of the article. And per reviewer’s suggestion, the authors have provided detailed experimental methods and conditions to improve the reproducibility of the experiment. In the revised manuscript, the authors emphasized that TTI can be used as an auxiliary strategy to expiration date to judge the quality change of food during storage, which can reduce the risk of consuming deteriorated food before the expiration and also help to reduce the food waste due to near the expiration date.
Response 2:
(1) The PP film was applied to collect the electrospun fibers. Related description has been modified in the “Materials and Methods” as:
The electrospun fibers were collected onto a polypropylene (PP) film which was attached to the drum collector, 12 cm from the tip of the needle.
(2) The SEM of electrospun chitosan fiber has been added as Figure 1 in the revised manuscript.
(3) The experimental conditions for immobilization of laccase have been stated in the material method of revised manuscript as:
To perform enzyme immobilization, 15 - 25 μg laccase (Sigma-Aldrich, from Trametes versicolor, ≥0.5 U/mg, lot result: 1.07 U/mg), in 50 μL buffer solution, was adsorbed carefully on a 1-cm2 piece of CS/PVA/TEOS/PP/GA, and the cross-linking reaction, by which covalent bond was formed between the amine group of laccases and the aldehyde group of GA, was set for 12 h at 5 °C to immobilize laccase on each piece of the film.
(4) The coloration skeme of laccase catalyze the oxidation guaiacol to form quinone and the chemical strucxcture of sodium azide have been added in the revised manuscript. The structure of laccase and guaiacol have been added in the Supplementary Material.
(5) The normalized absorbance (Absnorm) calculation for the color saturation of TTI has been cited the literature of Jayalakshmi and Santhakumaran (2011) in the revised manuscript.
(24. Jayalakshmi, T.; Santhakumaran, A. Statistical normalization and back propagation for classification. Int. J. Comput. theory eng. 2011, 3, 1793-8201.)
(6) Table 1 has been moved to "Supplementary material". The description of the endpoint of the coloration will refer to Figure 3 and Figure 4. The discussion of experimental results has been reinforced in revised manuscript.
(7) The multi-letter abbreviations in the “Materials and Methods” have been removed in revised manuscript.
Response 3:
Thanks to the reviewer's suggestion. The authors have rewritten the revised manuscript according to reviewer’s comments.

Reviewer 2 Report
The manuscript is technically sound and covers the scope of "Polymers".
This study can be considered for publication after addressing below
given suggestions;
1- In Abstract, Authors should mention a problematic statement. It seems
more like Conclusion, authors are advised to revise both Abstract and
the Conclusion. Conclusion contains significant values, and abstract
reveals important outcomes with a problem, novelty statement and results
in general to show potential for end uses.
2- Introduction contains long paragraphs, please divide into short
segments for authors ease.
3- Authors should include the discussion about importance and
applications of nanofibers within the introduction section with some
given reference;
[1] https://doi.org/10.3390/polym13081245
Also, add fibers used for food safety and packaging;
[1]https://doi.org/10.1557/s43579-020-00003-x
[2] https://doi.org/10.1080/10408398.2021.1899128
4- The text inside figures is small for readers, authors are advised to
increase the size of text used inside graphs.
Authors can only keep important graphs in the figures which are
optimized and directly related to applications, rest they can transfer
to supplementary file because graphs contain too much data, which should
be clear for readers and the graphs should be similar with respect to
style or appearance.
5- References are lacking in results and discussion section, authors are
advised to find the respective references and put at suitable places
within the manuscript. Next to reasoning statements.
Also, divide graphs into short segment rather than long single paragraph.
Best of Luck!
Author Response
The manuscript is technically sound and covers the scope of "Polymers".
This study can be considered for publication after addressing below
given suggestions;
1- In Abstract, Authors should mention a problematic statement. It seems more like Conclusion, authors are advised to revise both Abstract and the Conclusion. Conclusion contains significant values, and abstract reveals important outcomes with a problem, novelty statement and results in general to show potential for end uses.
Response 1:
The problematic statement (the novelty of the technology and the problems to be solved) has been added to the abstract, and the statement related to the value and application potential of this technology have also been reinforced in the conclusion of revised manuscript:
[Abstract]
The enzymatic time-temperature indicator (TTI) is usually suffered from its instability and inefficiency in practical use as food quality indicator during storage. The novelty of this study is to improve the aforementioned problem by immobilizing laccase on the electrospun chitosan fibers to increase stability and minimize usage of laccase. The addition of NaN3, as the enzyme inhibitor, was designed to extend this laccase TTI coloration rate and activation energy (Ea) range, so as to expand the application range of TTI for evaluating the quality changes of foods during storage.
[Conclusion]
The desirable tolerance and sensitivity of laccase TTI with NaN3 to temperature fluctuations is an important contribution of intelligent packaging to provide information on temperature alternation and time accumulation during food storage to reflect changes in food quality. This laccase TTI can be useful to monitor the quality change of foods during the storage, and to make up for the unreliability of judging the shelf life based on the expiration date on package.
2- Introduction contains long paragraphs, please divide into short segments for authors ease.
Response 2:
Long paragraphs have been divided into short segments to facilitate reading in revised manuscript.
3- Authors should include the discussion about importance and applications of nanofibers within the introduction section with some given reference;
[1] https://doi.org/10.3390/polym13081245
Also, add fibers used for food safety and packaging;
[1]https://doi.org/10.1557/s43579-020-00003-x
[2] https://doi.org/10.1080/10408398.2021.1899128
Response 3:
The literatures related to nanofibers have been cited in “Introduction” of revised manuscript:
Recently, bio-based electrospun nanofiber have gained substantial attention for preparing polymer-based biomaterials intended for use in cell culture [13], multifunctional silk fibroin-based devices [14], and food packaging [15].
- El-Ghazali, S.; Khatri, M.; Mehdi, M.; Kharaghani, D.; Tamada, Y.; Katagiri, A.; Kobayashi, S.; Kim, I.S. Fabrication of poly(ethylene-glycol 1,4-cyclohexane dimethylene-isosorbide-terephthalate) electrospun nanofiber mats for potential infiltration of fibroblast cells. Polymers2021, 13(8), 1245; https://doi.org/10.3390/polym13081245.
- Sun, H.; Marelli, B. Growing silk fibroin in advanced materials for food security. MRS Commun.2021,11, 31–45. https://doi.org/10.1557/s43579-020-00003-x.
- Sameen, D.E.; Ahmed, S.; Lu, R.; Li, R.; Dai, J.; Qin, W.; Zhang, Q.; Li, S.; Liu, Y. Electrospun nanofibers food packaging: trends and applications in food systems. Crit Rev Food Sci Nutr. 2021, 61 (on line). https://doi.org/10.1080/10408398.2021.1899128.
4- The text inside figures is small for readers, authors are advised to increase the size of text used inside graphs. Authors can only keep important graphs in the figures which are optimized and directly related to applications, rest they can transfer to supplementary file because graphs contain too much data, which should be clear for readers and the graphs should be similar with respect to style or appearance.
Response 4:
The Figures and Table in the text have been adjusted, including the Figures has been enlarged, the SEM of electrospun chitosan fibers has been added as Figure 1, and Table 1 has been moved to Supplementary material. More discussion of the experiment results and references have been added in revised manuscript.
32. Bobelyn, E.; Hertog, M.L.; Nicolaï, B.M. Applicability of an enzymatic time temperature integrator as a quality indicator for mushrooms in the distribution chain. Postharvest Biol. Technol. 2006, 42, 104-114.
38. Kim, M.J.; Shin, H.W.; Lee, S.J. A novel self-powered time-temperature integrator (TTI) using modified biofuel cell for food quality monitoring. Food Control 2016, 70, 167-173.
5- References are lacking in results and discussion section, authors are advised to find the respective references and put at suitable places within the manuscript. Next to reasoning statements.
Also, divide graphs into short segment rather than long single paragraph.
Response 5:
References and discussions have been added in the revised manuscript, and long paragraphs have been divided into short segment in “Introduction” and “Results and Discussion” to facilitate reading.
Reviewer 3 Report
My comments are the following:
- The abstract has to be rewritten since it is only describing the method and application. There is no aim at all.
- The possible application of the intelligent food packaging should be explained better in the Introduction part. The following reference should be included: Jancikova, S., Dordevic, D., Tesikova, K., Antonic, B., & Tremlova, B. (2021). Active Edible Films Fortified with Natural Extracts: Case Study with Fresh-Cut Apple Pieces. Membranes, 11(9), 684.
- The aim is not clearly explained at the end of the Introduction part.
- Principal component analysis should be performed.
Otherwise, the manuscript is explaining and dealing with the important aspect of food science. It is written in good scientific way.
Author Response
- The abstract has to be rewritten since it is only describing the method and application. There is no aim at all.
Response 1:
The problematic statement (the novelty of the technology and the problems to be solved) and main research methods have been added to the abstract in revised manuscript:
The enzymatic time-temperature indicator (TTI) is usually suffered from its instability and inefficiency in practical use as food quality indicator during storage. The novelty of this study is to improve the aforementioned problem by immobilizing laccase on the electrospun chitosan fibers to increase stability and minimize usage of laccase. The addition of NaN3, as the enzyme inhibitor, was designed to extend this laccase TTI coloration rate and activation energy (Ea) range, so as to expand the application range of TTI for evaluating the quality changes of foods during storage.
- The possible application of the intelligent food packaging should be explained better in the Introduction part. The following reference should be included: Jancikova, S., Dordevic, D., Tesikova, K., Antonic, B., & Tremlova, B. (2021). Active Edible Films Fortified with Natural Extracts: Case Study with Fresh-Cut Apple Pieces. Membranes11(9), 684.
Response 2:
The discussion of intelligent food packaging related literature has been added in “Introduction” of the revised manuscript:
Fresh-cut fruits and vegetables (FCFV) in the cold chain is prone to deterioration or microbial contamination. Although the use of active packaging, such as coating the edible film with natural polyphenols on freshly cut apple pieces, can increase the antioxidant activity of FCFV [23], most of the quality of them cannot be judged from the appearance. The TTI, prepared by immobilizing laccase on electrospun chitosan fibers, has been proved to be useful in monitoring the quality change of FCFV in the cold chain [17].
23. Jancikova, S., Dordevic, D., Tesikova, K., Antonic, B., & Tremlova, B. Active Edible Films Fortified with Natural Extracts: Case Study with Fresh-Cut Apple Pieces. Membranes 2021, 11(9), 684-697. (https://doi.org/10.3390/membranes11090684)
- The aim is not clearly explained at the end of the Introduction part.
Response 3:
The purpose of this study has been reinforced in the last paragraph of “Introduction” of the revised manuscript:
Fresh-cut fruits and vegetables (FCFV) in the cold chain is prone to deterioration or microbial contamination. Although the use of active packaging, such as coating the edible film with natural polyphenols on freshly cut apple pieces, can increase the antioxidant activity of FCFV [23], most of the quality of them cannot be judged from the appearance. The TTI, prepared by immobilizing laccase on electrospun chitosan fibers, has been proved to be useful in monitoring the quality change of FCFV in the cold chain [17]. In the present study, the coloration rate and Ea of the laccase TTI and inhibitory kinetic of NaN3 were investigated to extend the Ea range and coloration duration. Moreover, adjustments to the quantities of laccase and NaN3 was designed to expand the range of application of laccase TTI in intelligent packaging for food products in order to improve the accuracy of the TTI to estimate the quality changes of various foods.
- Principal component analysis should be performed.
Response 4:
Thanks to the reviewer's suggestion. This research mainly focused on the effect of adding sodium azide to TTI on the activation energy and the change of the coloration rate during isothermal or temperature fluctuations storage. The response test to the storage quality of FCFV and PCA analysis will be presented in our another research paper.
Reviewer 4 Report
The manuscript submitted to Polymers drives the hypothesis of the application of innovative laccase TTI on intelligent packaging by adding enzyme inhibitor to change its coloration kinetics. The manuscript is a novel contribution to the field and data are exhaustively presented. My only concern is that the authors did not apply these research findings in real food samples that were packaged in specific films. It could be of great novelty to have data on foods. In addition, for which foods are the authors confident that this will work?
Some other comments to improve the article:
-Statistical analysis section
The lines 180-182: These lines must be improved: The average values were compared using ANOVA at the least significance level p<0.05''.
-Line 251. Change ''resting'' to ''rest'' or ''other'', i.e. ''other form...''.
Based on the aforementioned, I leave the final decision to the Editor of the Journal.
Author Response
The manuscript submitted to Polymers drives the hypothesis of the application of innovative laccase TTI on intelligent packaging by adding enzyme inhibitor to change its coloration kinetics. The manuscript is a novel contribution to the field and data are exhaustively presented. My only concern is that the authors did not apply these research findings in real food samples that were packaged in specific films. It could be of great novelty to have data on foods. In addition, for which foods are the authors confident that this will work?
Response 1:
The authors have applied this TTI to monitor the quality changes of fresh-cut fruits and vegetables (FCFV) during storage and revealed that this TTI can effectively respond to the storage quality of FCFV. However, this research containing a lot of experiments, figures, tables data and text, and is not likely to be included in this manuscript. These research results will be presented in our another research paper. The application of laccase TTI and purpose of this study has been reinforced in the last paragraph of “Introduction” of the revised manuscript:
Fresh-cut fruits and vegetables (FCFV) in the cold chain is prone to deterioration or microbial contamination. Although the use of active packaging, such as coating the edible film with natural polyphenols on freshly cut apple pieces, can increase the antioxidant activity of FCFV [23], most of the quality of them cannot be judged from the appearance. The TTI, prepared by immobilizing laccase on electrospun chitosan fibers, has been proved to be useful in monitoring the quality change of FCFV in the cold chain [17]. In the present study, the coloration rate and Ea of the laccase TTI and inhibitory kinetic of NaN3 were investigated to extend the Ea range and coloration duration. Moreover, adjustments to the quantities of laccase and NaN3 was designed to expand the range of application of laccase TTI in intelligent packaging for food products in order to improve the accuracy of the TTI to estimate the quality changes of various foods.
Some other comments to improve the article:
-Statistical analysis section
The lines 180-182: These lines must be improved: The average values were compared using ANOVA at the least significance level p<0.05''.
Response 2:
The description of statistical analysis has been modified in revised manuscript as:
The average values were compared using ANOVA at the least significance level p<0.05.
-Line 251. Change ''resting'' to ''rest'' or ''other'', i.e. ''other form...''.
Response 3:
This sentence has been modified in revised manuscript as:
In the other form of the fungal laccases, …
Round 2
Reviewer 1 Report
This version has a lot of improvements in comparison to the previous one. Some problems are not solved correctly, yet overall it's almost ready to be published.
Multi-letter jargon descriptions of the materials produced are still present, even these descriptions are obsolete (like tetraethoxysilane described as TEOS, despite chemical description Si(OEt)4 is far better and commonly known), or CS/PVA/TEOS/PP/GA/laccase, suggesting that GA was connecting to PP (not true), as it was used as a spacer between chitosan and laccase. Laccase/GA/CS-PVA-Si(OEt)4/PP would look far better, yet it's still a jargon type description. It would be far better to use coding, e.g., M1 (material 1), or M1-laccase, even TTI -laccase looks far better than very hard-to-read multi-letter jargon description.
Could you, please, explain the connection between Ref. 24 and your methodology? The mentioned paper treats about use of neural networks in Diabetes Mellitus studies, not coloration.
Author Response
Multi-letter jargon descriptions of the materials produced are still present, even these descriptions are obsolete (like tetraethoxysilane described as TEOS, despite chemical description Si(OEt)4 is far better and commonly known), or CS/PVA/TEOS/PP/GA/laccase, suggesting that GA was connecting to PP (not true), as it was used as a spacer between chitosan and laccase. Laccase/GA/CS-PVA-Si(OEt)4/PP would look far better, yet it's still a jargon type description. It would be far better to use coding, e.g., M1 (material 1), or M1-laccase, even TTI -laccase looks far better than very hard-to-read multi-letter jargon description.
Response 1:
(1) The The abbreviation of tetraethoxysilane has been altered to Si(OEt)4 in the revised manuscript.
(2) The multi-letter jargon description has been substituted by the material coding in the revised manuscript:
2.1. Immobilizing laccase on electrospun fibers
Following the method of Tsai et al. [12], a chitosan (CS, DD: 76%, 50–190 kDa, Sigma-Aldrich, St. Louis, MO, USA) composite gel solution with polyvinyl alcohol (PVA, 118–124 kDa, First Chemical Manufacture, Taipei, Taiwan), and tetraethoxysilane (Si(OEt)4, Sigma-Aldrich) was electrospun on a polypropylene (PP) film with an electrospinning apparatus (NE-300, Falco Enterprise, New Taipei City, Taiwan). The electrospun fibers were collected onto a polypropylene (PP) film which was attached to the drum collector, 12 cm from the tip of the needle. The completed single-side electrospun film (material 1, M1) was soaked in 3% glutaraldehyde (GA; Nihon Shiyaku Industries, Osaka, Japan) and incubated for 2 h. Subsequently, the GA-modified M1 was washed with acetate buffer (pH 4.5) and dried overnight in a desiccator at ambient temperature to produce M2. A total of three sheets of M2 were produced, and 20 pieces each measuring 1 cm2 were cut from the middle section of each sheet. A minimum of three pieces were randomly selected for enzyme immobilization and coloration tests. To perform enzyme immobilization, 15 - 25 μg laccase (Sigma-Aldrich, from Trametes versicolor, ≥0.5 U/mg, lot result: 1.07 U/mg), in 50 μL buffer solution, was adsorbed carefully on a 1-cm2 piece of M2, and the cross-linking reaction, by which covalent bond was formed between the amine group of laccases and the aldehyde group of GA, was set for 12 h at 5 °C to immobilize laccase on each piece of the film. The electrospun film–immobilized enzyme was dried in an oven for 1 h at ambient temperature to produce M2-laccase. In addition, the occurrence of a coupling reaction in M2-laccase was tested by rinsing it with a guaiacol solution to verify that the laccase was effectively immobilized.
2.3. Coloration of laccase TTI prototype
The 1-cm2 pieces of M2-laccase with varying amounts of immobilized laccase (15–25 μg/cm2) were immersed in 1 mL of N-HGG solution to obtain the laccase TTI prototype and to investigate its coloration.
Could you, please, explain the connection between Ref. 24 and your methodology? The mentioned paper treats about use of neural networks in Diabetes Mellitus studies, not coloration.
Response 2:
The Ref. 24 discussed various types of “Normalization Procedures” and elaborated that the different techniques can use different rules such as Z-Score, max rule, min rule, sum rule, product rule and so on. The normalization of TTI coloration (OD500) in our study referred to the Z-score normalization method after comparing the applicability of these normalization techniques.
Reviewer 4 Report
The authors have responded effectively to my comments. Therefore, I suggest the publication of their study. Their future study will provide additional findings in the literature.
Author Response
Reviewer-4-2
The authors have responded effectively to my comments. Therefore, I suggest the publication of their study. Their future study will provide additional findings in the literature.
Response 1:
The authors are thankful for the reviewer's valuable suggestions on this manuscript.